# A Semiotic Reading of Aron Gurwitsch's Transcendental Phenomenology

Simone Aurora

Department of Philosophy, Sociology, Education and Applied Psychology (FISPPA), University of Padova, 35122 Padova, Italy; simone.aurora@unipd.it

**Abstract:** The aim of the paper is to show the relevancy of Aron Gurwitsch's transcendental-phenomenological theory of the field of consciousness for semiotics and the theory of meaning. After a brief biographical introduction, the paper will focus upon the key theoretical points that define Gurwitsch's theory of the field of consciousness and will consider some of Gurwitsch's reflections on linguistic and semiotic issues. Finally, it will be shown that the latter are strictly connected with Gurwitsch's general philosophical framework and, accordingly, that it is possible (and fruitful) to provide a semiotic understanding of Gurwitsch's phenomenology.

**Keywords:** Gurwitsch; field; consciousness; context; relevancy

## 1. Biographical Introduction: Between Phenomenology and Gestalt Theory

Aron Gurwitsch was born in a Jewish family in Vilnius, Lithuania, in 1901. After the violent pogroms that took place in many cities of the Russian empire between 1903–1906, Gurwitsch's family had to move to Danzig, then part of the German empire. Here Gurwitsch attended the classical Gymnasium for twelve years, studying Greek, Latin, French, English, mathematics, and history. In 1919, Gurwitsch entered the university of Berlin, where he studied under the guidance of Carl Stumpf. He studied mathematics under Karatheodory, Schur, Schmidt, and Rademacher, physics under Max Planck, and philosophy under Riel, Erdmann, Dessoir, Hoffman, and obviously Stumpf. In a letter written to Dorion Cairns on the 27 April 1941, Gurwitsch claims:

> "Please believe me when I say that I feel highly indebted to the old German university and to men who taught there. I do not speak of Husserl [ . . . ] he was a genius such as appear once in centuries; it is a quite exceptional fortune to meet such a man. But I do speak of Stumpf, for instance" [1] (p. 96).

In this passage, one can find evidence of how Gurwitsch felt indebted to Stumpf, in addition to Husserl of course, whom Gurwitsch always considered as his real and sole master. If it is probably true that Stumpf's impact, as Lester Embree writes, "was more that of a teacher's guidance than of a thinker's influence" [2] (p. 42), it is nonetheless worth remembering that Carl Stumpf has been both one of the teachers of Husserl and the "father" of all the main figures who animated German Gestalt Psychology in the 20's: Max Wertheimer, Kurt Koffka, and Wolfgang Köhler [1].

This circumstance should not be, in my opinion, underestimated, since Gurwitsch defines his philosophical project, already in his 1928 dissertation, as a study "of the relation between gestalt theory and phenomenology", as the subtitle of the dissertation reads [Studien über Beziehungen von Gestalttheorie und Phänomenologie] [4].

After two years in Berlin, following Stumpf's advice, Gurwitsch moved to Freiburg, where Husserl was teaching. Although personal contacts between Gurwitsch and Husserl did not take place until 1928, Gurwitsch's impression regarding Husserl's work was immediately enormous, so much so that the young student took the decision, as Gurwitsch himself writes,

"to devote his life and work to the continuation and expansion of Husserl's phenomenology—in a word, to remain a disciple forever, faithful to Husserl's spirit and general orientation, but at the same time prepared to depart from particular theories if compelled to do so by the nature of the problems and the logic of the theoretical situation". [5] (pp. XV–XVI)

After one year spent in Freiburg, Gurwitsch moved to Frankfurt to study with the Gestalt psychologists Kurt Goldstein and Adhémar Gelb, who were working on the psychological effects of brain injuries, with a focus on the phenomena of amnestic aphasia. In 1928, he defended his doctoral dissertation, *Phänomenologie der Thematik und des reinen Ichs. Studien über Beziehungen von Gestalttheorie und Phänomenologie*, which was published in 1929 on *Psycologische Forschung*, a journal founded in 1921 by Wertheimer, Koffka, and Köhler that became the official journal of German Gestalt psychology. The dissertation was sent to Husserl and became the occasion of personal meetings with him and some of his students, such as Dorion Cairns, Eugen Fink, and Ludwig Landgrebe. Gurwitsch came back to Berlin, where he received a *stipendium* to write his *Habilitation* thesis, but in April 1933, he and his wife Alice had to leave Germany because of the new government of Adolf Hitler. They went to France, where Gurwitsch taught at the Institute d'Histoire des Sciences et des Techniques at the Sorbonne. These courses dealt mainly with psychological issues and, especially, with Gestalt psychology and had great influence on Merleau-Ponty, who attended them. More generally, Gurwitsch's teaching in France was quite important for the reception of Husserl's phenomenology in the country. Sartre's theory of the "Transcendence of the Ego", for instance, follows the lines of Gurwitsch's revision of Husserl's phenomenology which was developed in his dissertation, as Gurwitsch himself acknowledges.

Finally, in 1940, Gurwitsch had to move again after the German occupation of France and emigrated with his wife in the USA, where he obtained various teaching positions before becoming professor at the New School for Social Research in 1959, when he took the place of his close friend Alfred Schutz, who had just died. In 1969, a Husserl Archive was established at the New School and Gurwitsch became the chairman of the board of directors. Moreover, in the USA, Gurwitsch wrote his main work, first published in 1957 as a French translation with the title *Théorie du champ de la conscience* [2].

This extended presentation of Gurwitsch's biographical and intellectual trajectory, which may appear somehow exaggerated, serves different purposes: first of all, it provides information for gaining insight into the scientific background of a marginal figure, as Aron Gurwitsch seems to be when compared with phenomenologists such as Husserl, Heidegger, Sartre, or Merleau-Ponty; moreover, it allows for situating Gurwitsch's philosophical and scientific enterprise at the intersection between phenomenology and Gestalt theory, as his relationship with Husserl, on the one hand, and Stumpf and the Gestalt theorists, on the other hand, clearly shows; finally, and against this background, it helps understanding why there is no explicit semiotic or linguistic theory in Gurwitch's work.

If we can assume that "semiotics is the knowledge developed by studying the action of signs and all that that action involves, including codes" and that "the action of signs as such springs from the being of signs as triadic relations" [6] (p. 49), we can easily appreciate its very general nature. However, since "a sign—any sign—is a sign by virtue of a relation irreducibly triadic attaining that which it signifies directly and an interpretant indirectly as its 'proper significant outcome'" [6] (pp. 47–48), we can say that sign and semiosis cannot be considered as primary elements, because they depend—as animal products—on the structure of a non-animal "triadic" relationship. It is the structure of this non-animal, namely transcendental, triadic relationship underpinning semiosis that occupies the focus of Gurwitsch's transcendental-phenomenological framework, as the triadic form of the field shows. In other terms, we cannot find any explicit semiotic theory in Gurwitsch's work because its formal theory of organization is intended to be more general than any other scientific theory: in Husserl's wording, Gurwitsch's formal theory of organization must be understood as the theory of all possible theories, semiotics included.

In what follows, I will try to explain the sense in which this theory of organization aims to uncover the conditions of possibility of semiosis, providing, in this way, some fundamental elements for a discussion about the nature of semiotics and of its main purposes. To this end, it is first necessary to present the key features of Gurwitsch's theory of the field of consciousness as a formal theory of organization.

## 2. The Theory of the Field of Consciousness

The originality of Gurwitch's position lies in the transcendental generalization of the insights of Gestalt theory, which are extended far beyond the domain of perception to the dimension of experience and are then explicitly integrated into a phenomenological framework. One of the cornerstones of the theory of the field of consciousness is represented by what Gurwitsch names the "principle of organization", according to which organization must be understood as an autochthonous feature of experience. According to this principle, organization must be understood as "an autochthonous feature of experience" [7] (p. 28) in so far as "organization emerges out of the experiential stream and thus proves a feature immanent to, and exhibited by immediate experience, not bestowed upon the latter from without" [7] (p. 32). Thus, the theory of the field of consciousness aims at providing a thorough description of the structural inner organization of experience.

> "Our aim is not to enumerate several possible types of organizational forms or to point out the qualitative differences between them. Rather we venture to assert the existence of a universal, formal pattern of organization, realized in every field of consciousness regardless of content. Every field of consciousness comprises three domains or, so to speak, extends in three dimensions. First, the theme: that with which the subject is dealing, which at the given moment occupies the "focus" of his attention, engrosses his mind, and upon which his mental activity concentrates. Secondly, the thematic field which we define as the totality of facts, copresent with the theme, which are experienced as having material relevancy or pertinence to the theme. In the third place, the margin comprises facts which are merely copresent with the theme but have no material relevancy to it". [7] (p. 53)

Let us consider in detail each one of these three elements in the following sub-sections.

### 2.1. The Theme

In the 1929 article, we find this definition of theme:

> "Without exception, in what follows we shall designate with "theme" that which is given to consciousness, precisely just as and only to the extent to which it is given and as it is disclosed by a strictly descriptive analysis. When we speak of the theme of an act of consciousness, we mean, accordingly, the "object" as it stands before our mind, as it is meant and intended through the act in question. In the present essay, the term "theme" has a noematic meaning, and a phenomenology of the theme is a noematic analysis". [8] (p. 202)

As Gurwitsch clearly states using a Husserlian wording, the theme is nothing but the noema. It is not the object of my conscious experience, since in conscious experience I am not faced with objects, but only with partial and perspectival sides of the object. Let us say that my theme, at the moment, is the paper I am writing. The paper is a complex theme and yet manifests itself as a unitary noema. What allows the theme to present itself as unitary, although it seems complex in nature? Here, Gurwitsch makes explicit use of Gestalt laws and defines the principle of organization of the theme in terms of Gestalt coherence.

> *There is no unifying principle or agency over and above the parts or constituents which coexist in the relationship of mutually demanding and supporting each other. The Gestalt, the whole of Gestalt-character is the system, having internal unification of the functional significances of its constituents; it is the balanced and equilibrated belonging and functioning together of the parts, the functional tissue which the parts form; more correctly, in which they exist in their interdependence and interdetermination. The unity*

*of the theme thus proves unity by Gestalt-coherence throughout, entirely and exclusively. Upon establishing the concept of Gestalt-coherence, we set forth the first dimension of conjunctions or the first formal type of organization".* [7] (p. 135)

Between the components of the theme, there obtains a relation of Gestalt coherence; this means that each component of the theme has a functional significance:

"The integration of a constituent into a whole of Gestalt character purports absorption of the constituent into the organizational structure of the whole. To be a constituent and, in this sense, a part of a Gestalt means to exist at a certain place within the structure of the whole and to occupy a certain locus in the organization of the Gestalt. The locus can be defined only with reference to, and from the point of view of, the topography of the whole. By virtue of its absorption into the structure and organization of a Gestalt contexture, the constituent in question is endowed with a functional significance for that contexture". [7] (p. 112)

So, each part of the paper, as my theme, has a functional significance, and accordingly, the paper as I experience it and as the reader might experience it represents a unity by Gestalt coherence.

The theme, however, never manifests itself as isolated, but always es emerging from a field, which Gurwitsch calls a "thematic field" and which represents the second level of organization of the field of consciousness. Before describing this though, we need to better clarify the notion of *functional significance*, which will prove to be fundamental for a semiotic interpretation of Gurwitsch's theory as a whole.

Functional Significance

In *Théorie du champ de la conscience*, one can find the same description of the structure of the field of consciousness based on a three-layered model already exposed in the 1929 article. In the 1957 book, there are nonetheless some important conceptual integrations. Concerning the theme, the most important element of innovation is the notion of *functional significance*. As we have already seen, according to Gurwitsch the "force" binding together the elements of the theme must be described in terms of a coherence of Gestalt nature. Gurwitsch describes "gestalt coherence" as *the determining and conditioning of the constituents upon each other*" [7] (p. 131). It is in this context that Gurwitsch introduces the notion of "functional significance": "*In thoroughgoing reciprocity, the constituents assign to, and derive from, one another the functional significance* which gives to each one its qualification in a concrete case" [7] (p. 131). In other terms, the elements forming the theme assume a value and a meaning only through their being related with all the other elements which define the boundary of the theme. In this way, we also gain a first definition of meaning: meaning is nothing but the function performed by a member of a whole with regard to the whole itself and to the other members of the whole.

There are two more conceptual elements that need to be considered.

If we assume the "principle of auto-organization of experience", we can claim that functional significance is not something bestowed or imposed by virtue of a subjective agency; on the contrary, it represents "a genuine phenomenal character and must not be mistaken as secondary or supervenient" [7] (p. 113). The constitution of a theme is the result of an organizational texture which provides for the assignment of specific functional meanings to those elements which, in turn, constitute the theme itself. This means that, as Richard Zaner underlines, "there cannot be any question of priority between 'part' and 'whole'", and accordingly, "there cannot be any defensible dualistic account" [9] (p. XXXI).

We must now refer to the notion of "functional weight", which is strictly connected with that of "functional significance". "Functional weight "enables acknowledging the different elements of the theme to different degrees of importance: the functions performed by the different constituents of a theme are all essential for the definition of its structure, and yet, not all of them have the same "weight". As Gurwitsch observes,

"After a few notes of a piece of music have been played, a passage giving the decisive turn to the piece may occur. This decisive passage gives to the piece its definite character and physiognomy; the preceding notes have a rather prelusive significance". [7] (p. 128)

It is clear that each and every note is essential for the structuring of the piece of music, which is actually nothing but the totality of the functional significances assumed by its components. However, some of the notes mark crucial passages, which represent "salient" moments of the piece and which, accordingly, are supplied with a higher "functional weight".

### 2.2. The Thematic Field

Let us consider the second level of organization of the field of consciousness, namely the thematic field.

"The appearance of a theme must be described as emergence from a field in which the theme is located occupying the centre so that the field forms a background with respect to the theme. The theme carries a field along with it so as not to appear and be present to consciousness except as being in, and pointing to, the field. This, of course, does not mean that a given theme is indissolubly connected with only one field [ . . . ] It is only the formal type and structure of organization, the formal condition that every theme appears in, and refers to some field, which is an invariant of consciousness". [7] (p. 311)

The paper, as my theme, emerges from a field and stands on a background, for instance, the textual sources which underpin the argument. Of course, I could also use other literary sources, and in this case, the thematic field would be a different one. What is invariant, though, is the fact that each and every theme refers to a thematic field, which Gurwitsch also calls *context*. What links the theme to its thematic field and at the same time connects the different components of a thematic field is a relation that Gurwitsch names "unity by relevancy" or "unity by context".

"Besides being copresent with the theme, the data falling under [a thematic field] appear, moreover, as being of a certain concern to the theme. They have something to do with it; they are relevant to it. Here the relationship is not merely that of simultaneity in phenomenal time but is founded upon the material contents of both the theme and the copresent data. Such a relationship is intrinsic since it concerns that experienced together rather than the mere fact of its being experienced together. Items between which such an intrinsic relationship obtains do not merely coexist with each other; they are not merely juxtaposed. A unity with its own specific nature prevails between them. This unity exemplified by the appearance of any theme within its thematic field will be called unity by relevancy". [7] (p. 331)

To go on with my example, what belongs to the thematic field of my theme are not only the explicit references to texts and authors but also the quotations that I have in mind but that I did not put in the paper, for instance, the philosophy of perception of Merleau-Ponty, hypotheses regarding possible objections, and the like. They are not part of my theme and do not form a Gestalt coherence with it, since they do not have a functional significance within the theme.

The "thematic field" is a network of material relations, but this network is not a product of an arbitrary construction deriving from a subjective agency; on the contrary, it is the theme, in its specific determination, to constitute the plot of material relations that define its field.

"The theme, with which we are dealing, is inserted into this framework of sense. *Qua* theme, it has a special and privileged place; it is what we are concerned with, and the components of the thematic field are cogiven with the theme. This

distinction of the theme is decisive for the structure of the thematic field". [7] (p. 225)

The position of the theme necessarily implies the constitution of a thematic field that results from the material configuration of the theme itself. Everything which is included in the thematic field is, to some extent, relevant for the theme and pertinent to its potential specifications. Let us take, as a theme, the cup of coffee on the table in front of me; the thematic field of the cup of coffee can include, for instance, the kind of coffee, the material of which the cup is made, etc. These contents are not arbitrarily chosen. They are relevant and pertinent solely by virtue of the fact that the theme occupying the thematic field *is a cup of coffee*. In other words, it is the theme that defines the system of relevancies structuring the "geography" of the field. The thematic field, in turn, takes part in the definition of the identity of the theme, which changes every time the field changes: the theme $T$ occupying the center of the thematic field $F$, changes whenever $F$ changes in such a way as to become $F^1$, thus becoming $T^1$. Between the theme and the thematic field, there obtains a relation of bilateral non-independency of a Gestalt kind.

The Positional Index

If, in the case of the theme, the most important element of innovation to be found in *Théorie du champ de la conscience* was the notion of *functional significance*, and in the case of the thematic field, the most relevant innovation is the notion of "positional index". As Gurwitsch writes, "the thematic field confers a positional index upon the theme, a character indicating the position of the theme in the field. That position obviously depends upon the relations between the theme and items of the thematic field" [7] (p. 351).

Because it always and necessarily implies the "opening" of a field, the theme automatically receives a positional index deriving from the configuration of the field in which it is inserted. "The term positional index denotes whatever perspective, orientation, or characterization the thematic field bestows upon the theme" [7] (p. 352). In other terms, the thematic field outlines the "topography of the whole" [7] (p. 112): it is like a map containing all the possible places in which the theme can occur, and at the same time, it indicates the exact position in which the theme actually lies in the actual experience. However, it is essential to highlight that "the positional index does not contribute towards shaping and constituting the theme as to its material content", and accordingly,

"variations of the positional index still codetermines *the theme as experienced, as, in a given case, it stands before the experiencing subject's mind.* That depending upon, and changing along with, the thematic field, is not the theme itself, but the perspective under which the theme presents itself". [7] (pp. 352–353)

These words seem to be at variance with what Gurwitsch wrote in the 1929 article about the problem of thematic (or noematic) identity. There, Gurwitsch described the identity of the theme as something highly "fluid" and instable, in so far as a variation within the thematic field, no matter how big, would have dissolved the theme and implied its substitution with a new one. To overcome this seeming contradiction between the theoretical positions expressed in the two works at a distance of almost 30 years, we must consider a clarification conveniently offered by Gurwitsch:

"The variation of the thematic field relative to a given theme is limited by the condition that between the theme and the eventual thematic field the relation of relevancy, based upon the material contents of either, be preserved, of whatever kind and specific nature the relevancy might be in a given case". [7] (p. 354)

Correctly understood, this claim seems to solve the aforementioned contradiction, allowing for a logically coherent theoretical account of Gurwitsch's position. The variation of the thematic field can be of two different kinds: there can be a variation of the "positional index", namely, of the position of the theme within the field, or a variation of the material relations informing the field and binding it to the theme. In the first case, the theme is preserved and maintains its identity; in the second case, the theme loses its identity, and we

are faced with a different theme. We can consider again the cup example. I am staring at the cup of coffee on the table in front of me. The cup of coffee is my theme, and as such, it is included in a thematic field, whose items are the elements showing a material relevancy with the theme (the table's surface, the teaspoon, the sugar cubes, etc.). Let us imagine that the field undergoes a change. As we have seen, this change can refer to the positional index of the theme: I can move the cup or I can modify the position of the teaspoon; in this case, the "geography" of the field is altered, but this alteration is a mere modification of the positional index of the theme, because the relations of pertinence obtaining between the items of the field remain the same. Here the theme maintains its identity: it is the same theme, the same cup of coffee, only with a different positional index. On the contrary, when the change concerns the items of the field or the relations of pertinence between the theme and the thematic field, the theme loses its former identity, a new theme emerges and, with it, a new topography of the whole. If the very same cup of coffee in front of me is inserted into a radically different context, for instance, an Ethiopian coffee ceremony, we can say that, from a phenomenological point of view, *the cup is no more the same cup*, because its thematic–noematic identity has radically changed: we are in front of a new theme.

### *2.3. Marginal Consciousness*

As we have seen, the items belonging to the thematic field are not simply co-present with the theme but are relevant for the theme. Mere co-presence, instead, is what characterizes the last level of organization of the field of consciousness, which Gurwitsch calls marginal consciousness.

> "Whatever datum is experienced simultaneously with the theme, but does not relate to it through relevancy, falls into the margin which in our previous discussions has proven to be a domain of irrelevancy and mere copresence". [7] (p. 403)

Although it is a domain of irrelevancy and mere copresence, the marginal level is not completely devoid of organization. The relation that links the marginal level to the theme and thematic field is *simultaneity*. Thus, simultaneity is the principle of organization of marginal consciousness. Whatever manifests itself as not having a functional significance with the theme, nor a contextual relevancy, but is nonetheless simultaneously present, is a component of marginal consciousness. Therefore, not everything can enter the marginal consciousness, but only that which is simultaneously experienced. Marginal consciousness is the dimension of contingency. While writing my paper I am conscious of how I am dressed or how the room in which I am looks like, but these are marginal components of my conscious experience, which are merely experienced as co-present [3].

## 3. Context and Relevancy

We now have all the basic elements to explore the semiotic fruitfulness of Gurwitsch's transcendental-phenomenological account. To this purpose, we need to highlight the semiotic significance of two fundamental notions, which play, not by chance, a pivotal role in the phenomenology of Gurwitsch: context and relevancy [4].

### *3.1. Context: Meaning and Meaning-Field*

As Wendy L. Bowcher highlights, "a typical way of representing context is to model it paradigmatically, i.e., as a system network of options, which aims to achieve a description of the potential conditions of relevant contextual [ . . . ] relations for language [or any other code] in use, those conditions which have the potential to 'make a difference' to the language [or code] that is involved" [12] (p. 4). The concept of context thus proves to be essential for the formulation of the theory of the two axes of language, which was formulated by Kruszewski, Saussure, Jakobson, and Hjelmslev and occupied a central position in the frame of reference of structural linguistics and of semiotics in general. Notwithstanding the different views developed in various semiotic traditions, it is possible

to assume the basic features of this theory as a prerequisite of any semiotic analysis, since "all signs [ . . . ] enter into complex syntagmatic as well as paradigmatic contrasts and oppositions" [13] (p. 21). As Elmar Holenstein sums up,

> "the theory essentially states that every linguistic unit is extended along two axes. It appears in *combination* with other linguistic units that together form its *context* [ . . . ] On the other hand every unit and every group of units in a message represent a *selection* from a storehouse of units, the code, which can be *substituted* for the unit without making the message meaningless". [14] (p. 84)

Every linguistic unit and every sign cannot but manifest themselves as related to another linguistic sign or linguistic unit. In accordance with Husserl's inquiries in the first logical investigation, the sign is here understood as the "carrier of meaning" [6] (p. 253).

If we abide by Gurwitsch's theoretical framework, this claim finds a transcendental-phenomenological explanation. Insofar as we experience something as a sign and this becomes the object of our conscious life, we cannot but experience at the same time a plurality of items which are somehow connected to it and which form its "semiotic context" or, in phenomenological-transcendental terms, its "thematic field".

This semiotic interpretation of Gurwitsch's theory has been somehow encouraged by Gurwitsch himself when he maintains the existence of a strict correlation between theme, noema. and meaning. The theme, as we have seen, can be identified with the noema, and both belong to the realm of meaning.

> "It is not [...], at the moment of the subject's dwelling upon the proposition which is his theme, he had only an additional consciousness of other propositions, the former and the latter merely being simultaneously apprehended. On the contrary, in and through the pointing references, an intrinsic relationship is experienced between the theme and those other propositions. Given is not one proposition plus other propositions, but a meaning-field. Such a field consists of meanings and meaning-unities organized around, and with respect to, the theme, a meaning-unity itself". [7] (p. 317)

In compliance with the transcendental principles of organization of the field of consciousness, the appearance of a meaning implies the appearance of a meaning field, which depends upon the meaning and, in turn, exerts an influence on the meaning itself. Between meaning and the meaning field, there obtain the same relations that exist between theme and the thematic field.

If we apply the theory of the two axes of language to the theory of the field of consciousness, we can say that the experience of a meaning implies an operation of *combination* with other meaning unities that form its *context* or *meaning field*: in this sense, the relation between meaning and meaning field is a *syntagmatic* relation which is founded in what we could call the phenomenological-transcendental *syntax* of consciousness; On the other hand, the emergence of a theme derives from an operation of *selection* that allows for the possibility of *substitution* with other potential themes or meanings: in this sense, the relation between meanings is a *paradigmatic* relation which is founded in what we could call a phenomenological-transcendental *semantics* of consciousness.

Let us take the English word "car" whose meaning is definable as "a road vehicle with an engine, four wheels, and seats for a small number of people". As it is clear from this definition, the meaning of "car" is a complex unit, which comprises other meanings such as "engine", "wheel", etc. What gives this complex meaning a unitary structure is the kind of relation that binds together all its components: a Gestalt relation of *functional significance*. This means that each of these sub-meanings are necessary and cannot be removed without *changing or substituting the meaning*; indeed, a car without an engine and wheels would not properly be a car. However, the meaning expressed by the word "car" implies a variety of other meanings, such as, for instance "driver", "highway", "motorcycle", etc. These meanings form the meaning field of the meaning of "car", and this means that they are not strictly necessary for its definition but are still materially relevant to it. The meaning of "car"

emerges in combination with these other meanings, but the modification of one of these does not necessary lead to the substitution of the meaning theme. What gives the meaning field a unitary structure is the kind of relation that binds together all its components: *material relevancy*. It is apparent that the relation that links together the meanings of "car" and "highway" is very different from the one linking together the meanings of "car" and "fish". In terms of the theory of the field of consciousness, we can say that the meaning "fish" lies at the *margin* of the field of meaning whose center is occupied by the meaning "car". This means that the meaning "fish" is simply simultaneously co-present with the meaning "car", in the sense that it is virtually present in the same semiotic code, namely, the English linguistic system.

These examples may appear very different from what linguists would usually mention as examples of paradigmatic and syntagmatic relations, which are defined by the strict organization of linguistic syntax. However, I think that Gurwitsch's analyses still abide by the logic of paradigmatic-syntagmatic relations, as the following example offered by Saussure clearly suggests:

> "From the associative and syntagmatic viewpoint a linguistic unit is like a fixed part of a building, e.g. a column. On the one hand, the column has a certain relation to the architrave that it supports: the arrangements of the two units in space suggests the syntagmatic relation. On the other hand, if the column is Doric, it suggests a mental comparison of this style with others (Ionic, Corinthian, etc.) although none of these elements is present in space". [15] (pp. 123–124)

If we understand the notions of *combination* and *selection* in their transcendental meaning, we can comprehend why Gurwitsch's views can be considered as deeply related to the syntagmatic-paradigmatic logic. Indeed, *combination* and *selection* are transcendental operations of the field of consciousness, and as such, they provide the conditions of possibility of syntagmatic and paradigmatic relations. This argument presupposes an isomorphism between different kinds of wholeness, which Gurwitsch explicitly supports by combining the phenomenological concept of *whole* with the psychological idea of *Gestalt* and the linguistic notion of *structure* [5].

### 3.2. Relevancy: The Emergence of Meaning

We have seen that relevancy plays a crucial function in Gurwitsch's theoretical framework as well as in the attempt to give a semiotic interpretation to phenomenological philosophy, as Göran Sonesson has masterfully showed [6]. Although Lester Embree has stressed the existence of three kinds of relevancy in Gurwitsch's theory (see [11]), they all refer to a common general meaning. As Sonesson has correctly observed, this general meaning that Gurwitsch assigns to the term "relevancy" is the meaning of "depending on" or "*pertaining to* a particular domain" [17] (p. 32, italics mine). However, I think that one can find a more specific semiotic counterpart of this term, by referring to the phonological notion of "pertinence" [7], as Alfred Schutz explicitly encouraged Gurwitsch to do [8].

> "Pertinence can be defined as a rule of scientific description (or as a condition which a constructed semiotic object must satisfy) according to which, among the numerous determinations (or distinctive features) possible for an object, only those determinations which are necessary and sufficient in order to exhaust its definition must be taken into account. In this way, this object will not be confused with another of the same level, nor will it be overloaded with determinations which, in order to be discriminatory, are only to be taken up again on a hierarchically inferior plane [ ... ] In a less rigorous [ ... ] sense, by pertinence will be understood the deontic rule adopted by the semiotician, of describing the chosen object only from one point of view [ ... ] consequently retaining, with a view to the description, only the features that concern this point of view [ ... ] It is according to this principle that, in the first stage, for example, one will either

extract elements [ ... ] considered *relevant* for the analysis, or, on the contrary, eliminate what is judged *non-relevant*". [19] (p. 231, italics mine)

If this comparison holds true, why does Gurwitsch tends not to use the linguistic-semiotic term of "pertinence" instead of that of "relevance", despite his deep knowledge of structural linguistics and, more specifically, of the Prague school where the notion of "pertinence" was developed? [9] The answer to this question lies in the already stressed fundamental transcendental nature of Gurwitsch's theoretical framework. If "pertinence" must be understood as a descriptive rule of semiosis, "relevance" must be understood as its condition of possibility, namely, as a transcendental rule of experience. In other terms, "pertinence" is the semiotic-specific declination of the general structure of "relevance". That is why, in my opinion, Gurwitsch prefers to distinguish between the two terms, which belong to two different dimensions: while "pertinence" refers to the actual and "positive" dimension of semiosis, "relevance" refers to the essential (eidetic) and transcendental dimension of experience.

It follows that we can also apply what Gurwitsch writes about logic to semiotics as well as to any other scientific disciplines: so, just as in logic "*the phenomenon of context or pertinence in its unspecified or rather prespecified form is*, we submit, *a necessary condition of logical relationship in the proper sense*" [7] (p. 323). In semiotics, the phenomenon of context or pertinence in its unspecified or rather prespecified form, namely, relevance, is a necessary condition of a semiotic relationship in the proper sense.

> "One must [always] analyse particular perceptual acts as well as the groups and systems into which the particular acts are interconcatenated. Correspondingly, the same questions can be raised concerning the constitution of "higher" universes, such as those of science, logic, mathematics, [semiotics] etc. Again, one has to go back to the acts, the groups of acts, the specific operations and procedures of consciousness, in and through which the universes in question present and constitute themselves [ ... ] as those for which we take them in our conscious life" [21] (p. 402)

According to Gurwitsch, a science of signs, as any other science, must thus be rooted in a science of conscious experience.

To say of something experienced that is relevant means then saying that it depends on the theme and pertains to its particular domain. The relation of material relevancy obtaining between the theme and thematic field is a bit more complex, as I have already tried to show. As a matter of fact, the theme always implies the opening of a thematic field, but this can, in turn, exert an influence on the theme itself, to the point of implying its substitution with another theme. The point is weather the change in the thematic field concerns its components or the positional index assigned to the theme, namely, its place within the thematic field. From a semiotic point of view, we can say that we are faced with the problem of the influence of context on the value of signs and meanings. We will now consider how Gurwitsch's conceptual toolbar can be used to deal with this problem.

As we have seen, the meaning of "car" is first defined by the network of its functional components, "engine", wheels", "road" etc. This is the structural–functional nucleus of the meaning, and this is context-independent: this means that a "car" will always be a "road vehicle with an engine, four wheels, and seats for a small number of people". A car without these elements would not be a car but something else. However, the meaning of "car" can be influenced by its meaning field in a twofold way: if the meaning field implies a change in the positional index of the meaning theme "car", then the identity of the structural-functional nucleus of the meaning "car" remains the same; if, on the contrary, the items composing the meaning field are altered, then we have a radical modification of the meaning "car", to the point that its identity is completely destroyed.

Suppose that people stop using cars as a means of transportation and start using bikes or trains. In this case, the meaning of "car" would occupy a different position within the system of meanings composing its field, although its structural–functional nucleus

remains unaltered. In other terms, what is changed is its positional index, not its structural configuration: a car would remain a "road vehicle with an engine, four wheels, and seats for a small number of people". Let us take another example. Nowadays people tend to marry less than they used to do fifty years ago, and it is not unusual to hear people saying that "marriage has no more the meaning it had in the past". What this sentence properly claims is not that the definition of marriage is changed, but that its "positional index" is changed. Marriage is also nowadays "the state of being united as spouses in a consensual and contractual relationship recognized by law", but its position with respect to other social relations has been radically modified.

Suppose now that a "car" is found by a human community that does not use cars, nor have highways, nor know what an engine is, etc. In this case, the meaning field would completely change, since the community will give to this object the meaning, for instance, of a fixed abode. Thus, the meaning field of car will contain items such as "family", "sleep", "shelter", etc. This provokes a change in the structural-functional nucleus of the meaning "car", since now the meanings of "engine" and "wheels" have lost all their structural function. I this case, we are simply in front of a new object and a new meaning and not a bare modification of the meaning of "car".

Needless to say, a change in the marginal dimension of the field of meaning of "car" is completely irrelevant for the definition of the structure and positional index of it. Should something change in the meaning of the word "fish", nothing would happen to the meaning of the word "car".

The last thing that is worth stressing is that the relevance of something is not established by a subjective agency. This is one of the most interesting outcomes of Gurwitsch's reflection on relevancy, or better, on "systems of relevancies" that, in the vein of Gurwitsch, need to be, according to Sonesson, detached "in a sense, from the dependence on individual subject" [17] (p. 23). The individual subject has a marginal role in structuring the field of experience or the semiotic code in which they are inserted. Her role is reduced to taking a position within the field assuming, in this way, a function within the set of relations defining the field. I can choose to read a book, for instance, but I cannot exert any influence on the system of relevancies in which this book is inserted; I can even write a book but, also in this case, I can only choose to let this book assume the function that the system of relevancies will assign to it.

## 4. Conclusions

With this paper, I have tried to explore the possibility of using Gurwitsch's transcendental-phenomenological account within the field of semiotics and for a discussion of some of the crucial problems concerning the issue of meaning. Of course, things are much more complex than I had the possibility to show. The paper aims at giving a first input for a semiotic understanding of Gurwitsch's theory of the field of consciousness. This understanding would require a much more in-depth analysis and a survey of all the critical points, the complications, and the stretches that this phenomenological application to semiotic issues implies. Nevertheless, I do think that Gurwitsch's phenomenological framework can be developed in a fruitful way so as to interact with some of the crucial problems in semiotics and, moreover, that it is possible to think of a sort of "theory of the field of meaning".

**Funding:** This research received no external funding.

**Institutional Review Board Statement:** Not applicable.

**Informed Consent Statement:** Not applicable.

**Acknowledgments:** I am very grateful to the three reviewers of this paper and to the guest editors, professors Göran Sonesson and Jordan Zlatev, for their valuable suggestions and critical remarks, which have been essential for improving the strength of the argument and the quality of the paper. Of course, I take full responsibility for the contents of the paper.

**Conflicts of Interest:** The author declares no conflict of interest.

## Notes

1    It is important to highlight that it is precisely under the guidance of Stumpf that Gurwitsch comes to conceive of an explicit integration between Husserlian phenomenology and Gestalt theory. In his classic study, Herbert Spiegelberg had already aknowledged that "the decisive reason for giving Stumpf as prominent a place [ . . . ] is the role he played in introducing phenomenological methods into psychology and trasmitting them to some of its most active researchers. In particular, Stumpf's approach permeated the work of the gestaltists (chiefly through Wolfgang Köhler, Max Wertheimer, and Kurt Koffka), the Group Dynamics movement (through Kurt Lewin), and, indirectly, the new 'phenomenological psychology' of Donald Snygg and Arthur W. Combs" [3] (p. 54).

2    The editoiral history of this book is quite strange. Although originally written in English, it was first published in 1957 in a French translation authored by the poet, writer, and literary critic Michel Butor.

3    In *Marginal Consciousness*, a 150-page manuscript survived in Gurwitsch's *Nachlass* (folder C 28) and was posthumously published in 2010, Gurwitsch specifies "three sets of data"which are always present in the marginal section of the field, "whatever may be the theme of our mental activity": "1. A certain segment of the stream of consciousness. 2. Our embodied existence, and 3. A certain sector of our perceptual environment" [10] (p. 449).

4    For what concerns relevancy, it is worth recalling what Gurwitsch himself once told to Lester Embree: "Aron Gurwitsch once told the present writer that he considered his analysis of relevancy his greatest contribution" [11] (p. 205).

5    On the contrary, according to Sonesson, *structure* and *Gestalt* are two different kinds of wholes and must not be confused. See [16] (p. 87): "The notion of whole is itself ambiguous. Different notions of wholeness, viz. structure and configuration, as conceived by structural linguistics and Gestalt psychology respectively, are often confused [ . . . ] In both cases, to be more precise, the whole is really something more than its parts, as the Gestaltist saying goes, but in the structure it is the network of relations which is central, and the elements connected by the relations will thus appear to be more distinct from (though sometimes identical to) each other; in the configuration, however, the general idea of wholeness and of all the elements' belonging together predominates, and the elements themselves are only secondarily apprehended as separate parts (cf. Sonesson 1989: 81ff). Thus, in the configuration, the parts tend to disappear in favour of the whole; in the structure, it is the whole that impresses its properties on the parts".

6    The combination of phenomenology and semiotics has been one of the key features of Sonesson's scientific activity. For a recent contribution on the problem of relevancy also dealing with the phenomenology of Alfred Schütz and Aron Gurwitsch, see [17].

7    As Frédérique de Vignemont reminds, "Gurwitsch also qualified the relation of relevancy as a relation of pertinence" [18] (p. 137). According to Sonesson, Schutz's notion of "pertinence" is similar to the linguistic notion, whereas that of Gurwitsch is different. On the contrary, Sonesson argues that Gurwitsch's notion of relevancy is not totally different from that of Schutz, despite Gurwitsch's own opinion. Indeed, if it is true that Gurwitsch starts from the type and Schutz from the token, both have to take into account both token and type. For a general and rigorous comparison between Gurwitsch and Schutz on these issues, see [17].

8    "Alfred Schutz [ . . . ] urged Gurwitsch to use "pertinence" in English, but Gurwitsch uses "relevancy" almost always" [11] (p. 206).

9    Gurwitsch's acquaintance with Prague phonology is clearly attested by a long report of *Psychologie du Langage*, a thematic issue of *Journal de Psychologie Normale et Pathologique* featuring many of the most important linguists and semioticians of the time (Delacroix, Cassirer, Sechehaye, Bühler, Meillet, Vendryes, Brøndal, Trubetzkoy, Sapir, Jespersen, Bally, Gelb and Goldstein). In this report, Gurwitsch devotes not by chance a particular attention to the papers by Vendryes and Trubetzkoy. See [20] (pp. 427–432).

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
