# Peer review of "A Semiotic Reading of Aron Gurwitsch’s Transcendental Phenomenology"

_philosophies, doi:10.3390/philosophies8010001_

Round 1

Reviewer 1 Report

This paper aims to present the Aron Gurwitsch's thought as a contribution to Semiotics. The description of Gurwitch's ideas and positions is clear and well-documented, supported by several quotations, but there is a lack of clear anlysis and discussion about  its connection with Semiotics.

Section 1 and 2 (6 of 9 pages) is the description of Aron Gurwitsch's thought but without almost any reference to Semiotics. Section 3 and 4 (only 3 pages) are the sections with the relation with Semiotics is argued.

Section 3 begins with the comparison with structuralist approach (the two axes of language), which is a good point, but this analysis is vague and superficial. In addition, the paper doesn't go deeper into this analysis because no different semiotic approaches under the structuralist framework as discussed (such as Saussere's Semiology or Barthes' Semiology) are analysed in terms of Gurwitshc's ideas. This is a relevant drawback that undermines the paper contribution and originality.

Section 3.2 is more relevant. It presents a further elaboration of Gurwitsch's ideas in terms of Semiotics, illustrating them with several examples, but there is a lack of theoretical analysis supporting it.

I suggest, for improving the paper, expand section number 3 and reduce section 2. Semiotics theories should be elaborated more elaborated to make clear that semotic reading of Aaron Gurwitsch.

Author Response

I have highlighted in red all the changes and additions. I have extended section 3 and thus deepened the semiotic part of the paper. I have also added some methodological clarifications at the end of the introduction.

Reviewer 2 Report

First of all, let me say that this is valuable work. The author clearly has a deep understanding of the phenomenological work of Aron Gurwitsch. Even though his figure remains largely unacknowledged in contemporary phenomenological debates, the author has been able through all the pages to show the innovation and originality of this thinker. Gurwitsch has surely much to contribute to several contemporary debates so this manuscript surely represents a precious contribution. The author intriguingly draws a parallel between Gurwitsch's phenomenological work as developed in "The field of consciousness" and an eventual semiotic understanding of it, with a particular focus on language. I think the paper is totally worth to be published but there could be some formal, as much as, in terms of content possible improvements.

1) The introduction could develop a more precise methodological structure since, at the current states, focuses  almost only on biographical facts about Gurwitsch. While Gurwitsch's life and influences are interesting, do they really contribute to the scope of the paper? For example, for a non-expert reader, what is the benefit of knowing that Stumpf has deeply guided Gurwitsch? My ultimate suggestion is to get rid of the excessive autobiographical facts and instead replace them with an introduction that concerns the content of the paper.

2) The revised introduction would benefit by discussing why Gurwitsch's approach is innovative for current issues in philosophy of language and, on the other hand, what is meant with semiotic understanding. And more importantly, why is this dialogue important? How can these two traditions benefit one from the other? It would also be important to add a few lines that explain the order of the sections

3) I have no comments or corrections to make on the articulation of Gurwitsch's view. As before, the only change I would make is to contextualize better why the theme-thematic field-margin structure of the field of consciousness is useful to semiotics approaches to language.

4) The real changes I think should be made starting from section 3. I would reformulate the sections to explain what is meant, for a non-expert reader, what is meant with ‘semiotics’, what this tradition implies and what they aim for. Why is it necessary to understand language? Similarly important, should the claims of the authors be interpreted as arguing that Gurwitsch can be interpreted in semiotics terms or that Gurwitsch can offer something to semiotics?

5) Is the semiotics reading of Gurwitsch used to interpret semantics and pragmatics or syntax? It could be technically applicable to everything. Looking at the examples made, the author seems to be interested in semantics but it might be interesting to look anyway at Gurwitsch's work on logic and abstractions. I suggest the author read these papers to explain how we can go from an analysis of conscious experience to language and abstractions.

 Embree L. 2004. The Three Species of Relevancy in Gurwitsch//Gurwitsch Relevancy for

 Cognitive Science. Contributions to Phenomenology. Volume 52. Dordrecht, The

 Netherlands: Springer.

 In Gurwitsch's collection of essays "studies in phenomenology and psychology" also look at "Philosophical presuppositions of logic" and "On a percpetual root of abstraction"

6) It Would be nice if the author would extend the discussion on the phenomenon of context in semiotics.

7) I think the example about the car would benefit from the introduction of the concept of "order of existence" used by Gurwitsch to capture the natural groupings of consciousness mentioned at the end of the field of consciousness and in the essay "The problem of existence in constitutive phenomenology".

Ultimately, this paper represents an important attempt to connect Gurwitsch with current issues in semiotics, but I think that it could benefit from a stronger methodology, fewer biographical facts and a more intensive and introductory discussion on what semiotics is and how it relates to Gurwitsch. I am confident the author will manage to fix these small points

Author Response

I have highlighted in red all the changes and additions. I have extended section 3 and deepened the semiotic part of the paper (section 3). I have also added some methodological clarifications at the end of the introduction.

  1. I have tried to answer the questions of the reviewer and to better explain both the value of the intellectual-biographical presentation of Gurwitsch and of the structure of the argument of the paper. (point 2 of the reviewer)
  2. By expanding section 3, I think I answered (some of) the questions posed by the reviewer at the points 3. 4. 5., also by using (some of) the suggested bibliography.

Reviewer 3 Report

The article relates Aron Gurwitsch’s phenomenology to semiotic problems related to meaning. First, the author introduces Gurwitsch and his theory of the field of consciousness focusing on the structure of theme, field and margin and explaining the theory with examples of their own. They then apply the theory to the problem of meaning in context and to the syntagma/paradigma distinction.

This is a promising article. The exposition of Gurwitsch’s complex theory is for the most part clear, the comparison between his 1929 article and his later monograph is interesting, and the examples given are helpful. At the same time, the structure and argumentation of the paper are imbalanced, the contribution to scholarship is not made clear enough, and there are some technical problems. I therefore suggest major revision if the manuscript is to be accepted for publication. In the following, I will outline in more detail the issues I have in mind.

Structure and argumentation:

There is an imbalance between the part on Gurwitsch (6 pages, albeit including a lot of quoted material) and the section on semiotics (“Context and Relevancy”, less than 3 pages). The biographical sketch in particular seems overly detailed. None of this would be a problem if the contribution to semiotics were made clear from the start. In order to do so, the article should begin with an introductory section stating the theoretical problem or question (rather than with a “biographical introduction” of Gurwitsch), to which the section on semiotics would then explicitly give an answer (which in turn should also be taken up in the concluding section – the present “Conclusion” refers to “some of the crucial problems…” without naming them). At present, most of the references are somewhat vaguely to Gurwitsch’s “relevancy” or “fruitfulness” to semiotic problems which are only introduced very late in the paper. The abstract should be rewritten with the same goal in mind. Since both the paper and the abstract are relatively short, there would be enough space for doing this.

Contribution to scholarship:

The article aims to show the relevance of Gurwitsch’s theory to semiotic problems rather than merely at a semiotic “interpretation”, or “understanding” of it, as is stated in other places in the paper. But to fulfill this aim, the contribution to existing scholarship must be made clearer and supported by referring to the literature.

At present, the bibliography is rather short, especially as far as semiotics proper is concerned (only two references, which again reflects the imbalance I mentioned before). Again, this would not necessarily be a problem if those references were more thoroughly engaged. But Holenstein 2020 mainly seems to serve as a summary conduct for other theories (ll. 313ff.), and Sonesson 2018 (and earlier) has a lot more to say about the issues treated in the paper, including the relevance of Gurwitsch. Against this background, the claim that the paper gives “a first input for a semiotic understanding of Gurwitsch’s theory” (l. 442) is somewhat dubious. Similarly, that a “semiotic interpretation … has been somehow encouraged by Gurwitsch himself” (ll. 329f.) is a bit of an understatement, given that Gurwitsch, in his “Field of Consciousness”, already applies his phenomenology in detail to the meaning of words and sentences in context and the logical relations between then. Perhaps the author originally intended to go into more detail here, as they write in the abstract that they “will consider some of Gurwitsch’s reflections on linguistic and semiotic issues” (ll. 7f.), but this is not really done in the paper (the discussion/quotation in ll. 330-344 seems to brief to me). Both broader and deeper engagement with existing scholarship in semiotics and with Gurwitsch’s own “semiotic” considerations seems necessary to clarify the article’s contribution.

This would be an opportunity to show how, in the author’s view, Gurwitsch helps solve specific semiotic problems, or (depending on the author’s goal) to show in more detail the similarities and differences between a Gurwitschean phenomenology and modern semiotics / linguistics. One point worth exploring more might be the notion of “function” and “functional significance” involved in Gestalt theory and in (structural) semiotics – is it the same concept, or how is it different? Furthermore, the concept of “relevance” could be related to the differences between Gurwitsch and Alfred Schutz, also something treated in some detail both in Gurwitsch’s “Field of Consciousness” and in Sonesson 2018 (and earlier). An issue that in my view needs pursuing is the role of (inter-)subjectivity. Saying that “the noema … is not the object of my conscious experience” (l. 106) seems misleading, as Gurwitsch (as a phenomenologist) is of course talking about consciousness. While this becomes clearer later in the paper, I think more nuance is needed. To be sure, gestalt relations, relevance etc. are not established by “arbitrary” construction, by “secondary” imposition or by some free-floating “agency” of subjectivity, but they are still elements of subjectivity. That “systems of relevancies” are detached “in a sense, from the dependence on individual subject” (Sonesson 2018), does not mean, in my mind, that “the individual subject has no role in the structuring of the field of experience or of the semiotic code in which she is inserted” (ll. 430f.). The author’s example of writing a paper or a book seems to me to involve individual action or “agency” beyond merely letting the relevancies play themselves out, as seems to be implied, e.g., in ll. 435f. Again, Gurwitsch’s Husserlian ideas on the intersubjective character of “objectivity” or Sonesson’s discussion of Gurwitsch vs. Schutz seem pertinent here in a way that would help clarify Guriwtsch’s position and his semiotic significance.

Technical problems:

Spelling and grammar should definitely be checked throughout, including the spelling of proper names (such as Gurwitsch’s) and foreign (French and German) words, as well as the bibliography.

Citations and references must be double-checked. Some of the citations have inaccurate page numbers (l. 430) while others refer to the wrong item (l. 95: the reference should be to [6], not [5]).

Minor issues:

l. 16: Gdánsk/Danzig was not yet an exclave at the time.

l. 230: “…thematic theme” should read as “thematic field”.

Author Response

I have highlighted in red all the changes and additions. I have extended section 3 and thus deepened the semiotic part of the paper (section 3). I have also added some methodological clarifications at the end of the introduction.

I solved hopefully all the mistakes stated in the last two points (technical problems, minor issue), although in one case I saw after a check that it was not a mistake (the page number is correct) and in another case (Danzig) I change the term in order to avoid any possible misunderstanding.

The parts added in the introduction and in section 3, as well as the extension of the bibliography, were intended to face some of the critical points raised by the reviewer in the sections "Structure and argumentation" and "Contribution to scholarship".

Round 2

Reviewer 1 Report

The paper has followed the suggested comments and I think it should be accepted.

Some minor comments are:

- Splitting into two the last paragraph of section 1, it's too long a difficult to read.

- p. 9. The new text added should be split into two or three paragraphs, it's too long and difficult to read.

Author Response

I have divided the text in more paragraphs, as the reviewer rightly suggested.

Reviewer 3 Report

By considerably extending the introductory section and the section on “Context and Relevancy”, the author has to a great extent remedied the imbalance that I described in my review of the original submission and clarified what they argue to be Gurwitsch’s relevance to semiotics. Supported by additional footnotes and references, the revised article also engages more explicitly and more broadly with existing research.

I am not sure that when Alfred Schutz suggested that Gurwitsch’s notion of relevance should be termed “pertinence” to distinguish it from Schutz’s own “relevance”, he had phonology in mind, but the author certainly presents an interesting interpretation of these concepts. Also, the concerns I voiced about the unclarity of the subjective/objective status of relevance in the original submission remain even after the analysis of “pertinence” has been added. But I feel these are matters for discussion after the paper has been published rather than for the review process.

In sum, this is an interesting contribution. I would only suggest that the article and bibliography be spell-checked once again, including words from languages other than English as well as proper names. Aside from this minor request, I recommend publication.

Author Response

I have spell-checked the whole text, including bibliography.